# A New Approach for Motor Imagery Classification Based on Sorted Blind Source Separation, Continuous Wavelet Transform, and Convolutional Neural Network

**DOI:** 10.3390/s19204541

**Published:** 2019-10-18

**Authors:** César J. Ortiz-Echeverri, Sebastián Salazar-Colores, Juvenal Rodríguez-Reséndiz, Roberto A. Gómez-Loenzo

**Affiliations:** 1Facultad de Informática, Universidad Autónoma de Querétaro, C.P. 76230 Querétaro, Mexico; cortiz08@alumnos.uaq.mx (C.J.O.-E.); ssalazar05@alumnos.uaq.mx (S.S.-C.); 2Facultad de Ingniería, Universidad Autónoma de Querétaro, C.P. 76010 Querétaro, Mexico; rob@uaq.mx

**Keywords:** Brain-Computer Interface, Blind Source Separation, Movement Related Independent Component, Wavelet Transform, Convolutional Neural Network

## Abstract

Brain-Computer Interfaces (BCI) are systems that allow the interaction of people and devices on the grounds of brain activity. The noninvasive and most viable way to obtain such information is by using electroencephalography (EEG). However, these signals have a low signal-to-noise ratio, as well as a low spatial resolution. This work proposes a new method built from the combination of a Blind Source Separation (BSS) to obtain estimated independent components, a 2D representation of these component signals using the Continuous Wavelet Transform (CWT), and a classification stage using a Convolutional Neural Network (CNN) approach. A criterion based on the spectral correlation with a Movement Related Independent Component (MRIC) is used to sort the estimated sources by BSS, thus reducing the spatial variance. The experimental results of 94.66% using a k-fold cross validation are competitive with techniques recently reported in the state-of-the-art.

## 1. Introduction

The Brain-Computer Interface (BCI) is a method of communication between a user and a system, where the intention of the subject is translated into a control signal by classifying the specific pattern which is characteristic of the imagined task, for example, the movement of the hand and/or foot [1]. The most widely used technique to register the electrical activity for BCI applications is the electroencephalography (EEG), which is a non-invasive and low-cost technique. The recording is done by placing electrodes on the scalp according to the 10–20 system [2], which records electrical impulses associated with neuronal activity in the brain cortex. The BCI can be based on exogenous such as the event-related P300 potential and Visual Evoked Potentials (VEPs), or endogenous potentials, where Motor Imagery (MI) widely used in BCI applications is the dynamic state where a subject evokes a movement or gesture. The event related phenomena represent frequency-specific changes in the ongoing EEG activity and may consist, in general terms, of either decreases or increases of power in given frequency bands [1]. Most of the brain activity is concentrated in electrophysiological bands called: delta δ (0.5–4 Hz), theta θ (4–7.5 Hz), alpha α (8–13 Hz), and beta β (14–26 Hz) [2]. Another important frequency for applications in BCI is the μ or sensorimotor rhythm, with the same frequency bands as α, but located in the motor cortex instead of the visual cortex where α is mainly generated [3]. There are several works which report the importance of μ frequencies for MI detection [3,4,5,6,7], where Pfurtscheller et al. published [8,9,10,11,12]. They demonstrate the changes of EEG activity in μ and β rhythms caused by voluntary movements.

Endogenous MI-BCI-based system does not require external stimuli, hence it is more acceptable to the users [4]. Nonetheless, MI depends on the ability to control the electrophysiological activity, which makes feature extraction and classification for MI-BCI based system more difficult than for exogenous responses. One of the major limitations of EEG records is the low signal-to-noise ratio and the fact that the signals picked up at the electrodes are a mixture of sources that cannot be observed directly by non-invasive methods. Therefore, for endogenous BCI approaches, a preprocessing step is required to identify independent sources of the mixtures observed in the electrodes. A well-known preprocessing method is based on the decomposition of multi-channel EEG data into spatial patterns which are calculated from two classes of MI, known as Common Spatial Patterns (CSP) [13,14]. CSP is a supervised method where class information must be available a priori and its effectiveness relies on the subject-specific frequency bands [7,15].

As an unsupervised alternative to the estimation of independent sources, Blind Source Separation (BSS) algorithms have been incorporated in EEG preprocessing, mainly in medical applications to improve the tasks of diseases diagnosis [16]. BSS algorithms make the source estimations from the mixed observation using statistical information. It has been shown that BSS is especially suitable for removing a wide variety of artifacts in EEG recordings [17] and separating μ rhythms generated in both brain hemispheres [18]. Therefore, BSS is a useful method for constructing spatial filters for preprocessing raw multi-channel EEG data in BCI research [15].

Due to its unsupervised and statistical nature, BSS does not require a priori information about MI classes, nor specific frequency bands, which is an advantage over CSP approaches. Nonetheless, an inherent disadvantage of BSS algorithms is that for each processed trial, the order is not preserved, which limits its direct application in further classifier stages used in BCI, where the order of the input vectors must be conserved to avoid loss of the adjustment parameters for each new data entry. Some automated BSS approaches have been proposed to discern between sources of interest and artifacts, and thus minimize the aforementioned inconvenience making use of statistical concepts [19,20,21].

In the classification stage, the most widely used approaches are Linear Discriminant Analysis (LDA) [22], Support Vector Machine (SVM) [23], Multilayer Perceptron [24], and Bayesian classifier [25]. A recent approach that has given excellent results, mainly in computer vision is deep learning [26]. However, deep learning techniques have not been widely used for EEG-BCI applications, due to factors such as as noise, the correlation between channels, and the high dimensional EEG data [27]. Some works where deep learning has been used for MI classification have been proposed [27,28,29,30,31,32,33,34,35]. However, for MI-BCI based paradigm the datasets are small due to the fatigue where the participants are exposed in each session. Therefore, it has been difficult to use deep learning for this purpose [32].

In this research, a fastICA BSS algorithm is used to obtain estimated independent components. A typical spectral profile of Movement Related Independent Components (MRIC) with significant components in the μ and β frequencies is used for sorting in each processed trial, thus ensuring that the sources estimated to be the most active in MI frequencies remain at the beginning of the array, while the artifacts are placed in the final positions. For each estimated source, the Continuous Wavelet Transform (CWT) is calculated for a given time window, generating an image containing temporal, frequential, and spatial information. This process is carried out throughout all trials, forming a set of images to train and test a Convolutional Neural Network (CNN).

A contribution in the present work is the use of BSS instead of the widely used CSP. Even though this has worked well for MI-BCI based, these spatial filters require prior information of the classes to be separated in order to maximize the differences between them. In addition, BSS is an unsupervised approach that does not require prior information about the classes. The problem of large datasets needed for training is minimized using the MRIC criterion to sort the estimated sources. The paper is structured as follows: Section 2 the background of BSS, CWT, and CNN are explained. Section 3 the proposed methodology to obtain CWT maps from estimated sources is described, along with the details of the CNN architecture. The experimental results and discussion are presented in Section 4. Finally, conclusions and a future work overview are presented in Section 5.

## 2. Background

In this section, theoretical background about preprocessing, feature extraction, and classification stages are presented.

### 2.1. Blind Source Separation

Firstly, BSS is an approach to estimate and recover the original sources using only the information of the observed mixture at the recording channels. For the instantaneous and ideal case, where the source signals arrive to the sensors at the same time such as in the EEG, the mathematical model of BSS can be expressed as
(1)x(t)=As(t),
where x(t) is the vector of the mixed signals, A is the unknown non-singular mixing matrix, and s(t) is the vector of sources [2]. Then, BSS consists of estimating a separation matrix B such that in an ideal case $B=A^{-1}$, and then computed the sources $s^{\prime }\left(t\right)$ as
(2)$$s^{\prime }\left(t\right)=Bx\left(t\right),$$

The BSS algorithms are grouped into two main categories: (1) Second-Order Statistics (SOS), built from correlations and time-delayed windowed signals [36]; and (2) High Order Statistics (HOS) which are based on the optimization of relevant elements in the Probability Density Function (PDF) [37]. The prior SOS algorithms search for the independence of signals based on the criterion of correlation between them. However, uncorrelatedness does not imply independence in all cases. In the HOS algorithms, the assumption to find the estimated sources is that the independent sources have non-Gaussian PDF, while the mixtures present a Gaussian distribution, which is valid for most cases including EEG signals [38]. The classical algebraic approach of ICA is based on negentropy, which is a measure of the entropy of a random variable H(y) with respect to the entropy of a Gaussian distribution variable $H_{g}\left(y\right)$. By definition negentropy is the Kullback-Liebler divergence between a density of probability p(y) and the Gaussian probability density of g(y) of the same mean and same variance. The negentropy *J* of y is defined as
(3)$$J\left(y\right)=H_{g}\left(y\right)-H\left(y\right),$$

According to the definition of mutual information, I(y) can be expressed as
(4)$$I\left(y\right)=J\left(y\right)-\sum_{i}J(y_{i})+\frac{1}{2}\log\frac{\prod_{i}v_{ii}}{\det V},$$
where V is the variance-covariance matrix of y, with diagonal elements $v_{ii}$. Since maximizing independence means minimizing mutual information, maximizing of the sum of marginal negentropies is to minimize mutual information after whitening of the observations. This method is also similar to those using the notion of kurtosis. Under these conditions, the separation problem consists in the search for a rotation matrix J such that $\sum_{i}H\left(y_{i}\right)$ is minimal. The use of second-order moment is not sufficient to decide if non-Gaussian variables are independent. On the other hand, the use of cumulants (cross-referenced) of all kinds makes it possible to show whether variables are independent or not: if all the crossed cumulants of a set of random variables of all kinds are null, then the random variables are independent.

### 2.2. Wavelet Transform

Second, EEG registers are initially time series, from which it is possible to obtain information related to the temporal evolution of some characteristics. However, in this space, it is not possible to know frequency information. On the other hand, the Fourier transformation makes it possible to identify information in the frequency domain, but the temporal evolution of the frequency components is unknown. The CWT generates 2D maps from 1D time series containing information of time and scale, with a logarithmic relationship with the frequency components. Unlike the Short Time Fourier Transform (STFT), CWT performs a multiresolution analysis [39]. CWT is described by
(5)$$W_{x}\left(a,\tau \right)=\frac{1}{\sqrt{\left|a\right|}}\int_{-\infty }^{\infty }x\left(t\right)\psi^{*}\left(\frac{t-\tau }{a}\right)\;dt,$$
where *a* is the scale factor, ψ is the mother wavelet, and τ is the shift time of the mother wavelet on the x(t) signal.

### 2.3. Convolutional Neural Network

The Convolutional Neural Network (CNN) architecture is comprised of a sequence of convolutions and sub-sampling layers in which the content or values of the convolutional kernels is learned via an optimization algorithm. With a fixed number of filters in each layer, each individual filter is convoluted transversely with the width and height of the input figure in the forward transmits. The output of this layer is a two-dimensional feature map of that filter to detect the pattern. This is followed by a Rectified Linear Unit (ReLU) where the non-linearity is increased in the network using a rectified function. The governing equation of the convolution operation is given as
(6)$$h_{i}^{\ell }=f\left(c\right)=f\left(\sum_{n=1}^{M}W_{ni}^{\ell }\cdot h_{n}^{\ell -1},+b_{i}^{\ell }\right)$$
with the ReLU function is defined as
(7)$$f\left(c\right)=\left\{\begin{array}{cc} 0, & c<0,\\ c, & c\geq 0, \end{array}\right.$$
where $h_{i}^{\ell }$ is the *i*-th output of layer *ℓ*, $W_{ni}^{\ell }$ is the convolutional kernel that is operated on the *n*-th map of the ℓ−1-th layer used for the *i*-th output of the *ℓ*-th layer, and $b_{i}^{\ell }$ is the bias term; f(c) is an activation function imposed on the output of the convolution (c). The optimization algorithm focuses on optimizing the convolutional kernel *W*. The output of each convolutional operation or any operation is denoted as a feature map. After convolution, the sub-sampling layer reduces the dimension of the feature maps by representing a neighborhood of values in the feature map by a single value.

Additionally, CNN are also known as shift invariant or space invariant artificial neural networks (SIANN), based on their shared-weights architecture and translation invariance characteristics [40]. This particular feature makes CNN appropriate to deal with the problem described above with the loss of order in each trial processed by BSS algorithms. The parameters of each convolutional layer are:Filters: The number of output filters in the convolution.kernel size: The height and width of the 2D convolution window.Strides: The strides of the convolution along the height and width.

Another important operation applied usually in each convolutional layer is the max-pooling, that is a sample-based discretization process. The objective is to down-sample an input representation, reducing its dimensionality and allowing for assumptions to be made about features contained in the sub-regions binned. This is done to in part to help over-fitting by providing an abstracted form of the representation. Also, it reduces the computational cost by reducing the number of parameters to learn and provides basic translation invariance to the internal representation [41].

## 3. Methodology

In this section the dataset format, the selected electrodes, and the proposed approach for MI classification are described.

### 3.1. Dataset

The dataset was provided by Intelligent Data Analysis Group is the IVa from the BCI competition III was used in current study. This consists of five healthy subjects sat in a comfortable chair with arms resting. The subjects are labeled as *aa*, *al*, *av*, *aw*, and *ay* [42]. Visual cues indicated for 3.5 s which of the following two MI the subject should perform: right hand (class1) and foot (class2) MI. The presentation of target cues were intermitted by periods of random length, 1.75 to 2.25 s, in which the subject could relax [43]. In Figure 1 is shown a diagram where it is illustrated the extraction of MI segments inside a trial, where the interest segment of MI occurs in time interval between 4.5 and 8 s. The recording was made using BrainAmp amplifiers and a 128 channel Ag/AgCl electrode cap from ECI. 118 EEG channels were measured at positions of the extended international 10–20-system. Signals were digitized at 1000 Hz with 16 bit (0.1 μV) accuracy.

As mentioned above, μ rhythms are generated in the motor cortex [5], therefore, the selected electrodes for this region are $C_{3}$ for the left hemisphere and $C_{4}$ for the right hemisphere. In this work, 18 channels were located in both hemispheres around the sensorimotor cortex were selected. $FC_{5}$, $FC_{3}$, $FC_{1}$, $C_{5}$, $C_{3}$, $C_{1}$, $CP_{5}$, $CP_{3}$, $CP_{1}$ in left side, and $FC_{2}$, $FC_{4}$, $FC_{6}$, $C_{2}$, $C_{4}$, $C_{6}$, $CP_{2}$, $CP_{4}$,$CP_{6}$ in right side. The selected electrodes coincide with the regions reported in [21]. The two clusters left and right are filtered using a second-order IIR Butterworth, set as band-pass filter. To preserve the relevant information for MI, the selected cut-off frequencies were adjusted between 0.5 and 90 Hz.

### 3.2. Proposed Approach

Description of proposed approach is depicted in Figure 2. Clusters left and right are separated in trials, from $trial_{1}$ to $trial_{N}$. Each trial is preprocessed using a BSS algorithm, which generates equal number of estimated sources $s^{\prime }\left(t\right)$ from the input channels x(t). These sources were sorted using as criterion the correlation between their spectral components and the MRIC. This procedure helps to separate the sources and the unwanted artifacts that have low correlations with MRIC. Sorted trials of $s^{\prime }\left(t\right)$ are passed through a CWT block The CWT is obtained using generalized Morse wavelets. Analytic wavelets are complex-valued wavelets whose Fourier transforms are supported only on the positive real axis. They are useful for analyzing modulated signals, which are signals with time-varying amplitude and frequency [44]. The window size of each CWT is 1.0 s, each window is computed using steps of 0.25 s. CNN architecture has as input the CWT figures and finally a fully connected Multilayer Perceptron (MLP) separates into two classes, *right hand MI* and *right foot MI*. In Figure 3a is shown the CWT of a single estimated source $s_{1}^{\prime }$, in Figure 3b is shown an example of input containing all estimated sources stacked along *y* axis. Each figure is re-scaled to a size of 128×256.

An example of CNN architecture is depicted in Figure 4. Each CWT input images are passed through the CNN architecture. Two convolutional layers with respective max-pooling are responsible for obtaining descriptors from CWT maps. In the third layer, the matrices are flattened and passed through a dense layer. Finally, an output layer composed of two neurons is the classifier for two MI classes.

### 3.3. Experiment Setup

The experiments were conducted on an Intel Core(TM) i7-7700HQ 2.80 GHz with 16 GB of RAM. Matlab 2017 was used to compute fastICA [38] and CWT maps, while python Tensorflow and Keras libraries were used to compute the CNN architecture.

## 4. Results and Discussion

In this section a validation of the proposed method is made, some relevant hyper-parameters of the CNN architecture are explored, and the classification results are compared with other reported methods.

### 4.1. Validation of Proposed Method

The first experiment was carried out to compare the performance between two BSS approaches: SOS and HOS. To evaluate the proposed methodology, an SOS called Robust Second Order Blind Identification (SOBIRO), and the previously mentioned fastICA, belonging to the family of HOS algorithms were tested in one subject. In Figure 5 is showed the train and validation values for 20 epochs in cases Figure 5a without BSS preprocessing, Figure 5b with fastICA without the sort criterion, Figure 5c with sorted SOBIRO, and Figure 5d with sorted fastICA. Considering the structure of the database used in [32], 50% trials were set as the training set, and the remaining 50% were selected as the test set. The CNN architecture initially used was the same proposed in [32]. Nevertheless, the input data is organized in a different way, for which it was necessary to make some adjustments in the convolution stride and max-pooling size. The CNN architecture used in this work is showed in Table 1.

In the graphs it is possible to observe that in case (a) without preprocessing BSS stage, the maximum accuracy value for train is near 0.8, while the validation accuracy is below 0.6. These results can be explained taking into account the reduced number of data for a deep learning approach, where large amounts of data are required to achieve an end-to-end system, where convolutional layers are able to find the determining patterns that allow for classifying movement intentions; In the case (b), where fastICA is applied in each trial but without the sort step, the training accuracy reaches values close to 0.98, but the validation accuracy is below 0.60 and decrease for each epoch. This can be explained by the disorder of the estimated sources in each trial, where CNN learns training set, but fails to generalize in the test set. This result validates the hypothesis of the need to use a sort criterion for sources estimated through BSS; In case (c), using the second-order Statistic approach (SOS) SOBIRO as BSS algorithm and sorted with the same explained criterion in Figure 2, the test accuracy achieved 0.73 values, improving the (a) and (b) responses. However, differences between train and test are considerably large which indicates overfitting; Finally, in case (d), with the sorted HOS fastICA, both the training data and the validation data achieve values higher than 0.8, reducing the phenomenon of overfitting. This result validates several previous works where the superiority of the HOS algorithms over the SOS for BSS is reported. Results (c) and (d) are in accordance with numerous works reporting superiority of HOS-BSS algorithms over SOS-BSS algorithms for EEG preprocessing [45,46,47]. Sorted fastICA is then chosen as BSS algorithm before CWT generation and posterior CNN classification stages.

### 4.2. Comparison with Other Methods

In Table 2 are shown the validation accuracy for each *k* validation and each subject of dataset IVa of BCI competition III. The test set was divided into 10 equal parts for each cross validation. The maximum classification value was chosen in each case.

As is shown in Table 2, the average ranking percentage of 94.66%, with a standard deviation σ of 6.46. For the five subjects, the maximum k-fold accuracy average was 97.81% with σ of 3.34 in subject *aa*, and a minimum value of 92.18 with σ of 6.98 in subject *ay*. Table 3 shows some recent work that reports the same used dataset [48,49,50,51].

### 4.3. Discussion

One of the main contributions of this work is the criterion of sorting the estimated sources. In Figure 6 are shown the spectral components of estimated sources before and after apply the sort criterion in one trial. The components with more information in μ and β will generally be placed in the top positions, while the components least associated with these frequencies will be at the bottom as for example in Figure 6a, the first spectral component has more energy in frequencies over 20 Hz but without α and β contribution, is placed in the bottom in Figure 6b. In contrast, the component located in the seventh position in Figure 6a, which has the most energy in the region μ, is located in the first position in Figure 6b.

The other distinctive part of proposed method is the use of a CNN architecture, and the adjustment of some relevant hyper-parameters. Taking as initial values those shown in Table 1, and changing the kernel size along height and width for each convolutional layer, the behaviour of validation accuracy for each case is analyzed. The kernel size is (y,x), where *y*-axis contain frequential and spatial information, while *x*-axis contain temporal information. In Figure 7 are depicted the validation accuracy for subject *aa*. The kernel sizes selected to the comparison in the first convolutional layer were (i,1), (i,3), (i,5), (i,7), with *i* taking odd values from 1 to 9.

According to the analysis of kernel size in the first convolutional layer, the size (7,5) (Figure 7c) presented less overfitting and major classification accuracy. It is also noted that when the kernel has a size of in the size 3 for *y*-axis, while reaching a maximum value close to the best case, this kernel size generates a high overfitting after certain times. On the other hand, the the size 1 for *y*-axis generates the lowest maximum values in all combinations of *x*-axis.

These results are in accordance with other work where the size of the kernel is also studied, where they report that the vertical locations (frequency-space) is of great importance for the classification performance, while, in contrast, the horizontal locations (time) are not as significant [27].

Once fixed the kernel size in first layer, a similar analysis was made for the kernel of the second convolutional layer. In Figure 8 are depicted the validation accuracy for subject *aa*. Taking into account that a (4,4) max-pooling has been previously applied, the kernel sizes selected were (j,1), (j,2), (j,3), and (j,4), with *j* taking values from 1 to 5.

At this case, the only *y*-axis where the CNN can achieve a stable accuracy throughout the epochs is 1 independently of *x*-axis size. In other cases, the validation accuracy decreases to 60% or even around 50% in case *y*-axis = 5.

As is well known, currently deep learning approaches are state-of-the-art in many images processing and artificial vision. However, in contrast with two-dimensional static images, the EEG signals are dynamic time series, where the generalizable MI patterns in EEG signals are spatially distributed and mixed in the channels around the motor region. In addition to this, low signal-to-noise ratio could make learning features in an end-to-end fashion more difficult for EEG signals than for images [29]. On the other hand, deep learning approaches require a large amount of training data in order to obtain descriptors that allow discrimination between different classes. In particular case of EEG and MI, this is a limitation since the data must be processed independently for each subject, and due to fatigue, the MI databases are relatively small. To deal with this problem, some works proposed using deep learning have done data augmentation using some criteria to generate simulated data from the training set. This approach has yielded good results. However, the generation of artificial data can be risky without a rigorous methodology and thus generate false data that increases accuracy for a particular dataset.

## 5. Conclusions

In this work, the estimated independent components were obtained using a fastICA algorithm, separating relevant MI-related independent components from unwanted artifacts. However, the source separation by itself is not sufficient if for each trial, the order of them is not preserved. For this reason, a spectral correlation with MRIC helps to sort the sources by reducing the spatial variance, leaving in the last positions the sources with a more significant influence of artifacts and less μ and β components. These operations help to reduce the complexity in the search for relevant patterns in the posterior extraction and classification stages. The use of CWT maps in the feature extraction stage allows obtaining a 2D representation of time series. In contrast with Short-Time Fourier Transform (STFT), CWT performs a multi-resolution analysis. According to the experimentation carried out, obtained results in this work are competitive with the state-of-the-art with a 94.66% in the k-fold cross validation. Regarding the architecture of CNN, it was found that the hyper-parameters related to the size of the kernel as well as the kernel stride in each convolutional layer have a significant influence on network performance, while the number of convolutions has less impact in final accuracy. Two future works derived: first, the development of a methodology that allows to find the hyper-parameters close to the optimum and then, improve the current results. Second, to replace the BSS stage with some autoencoder architecture, as for example Variational Autoencoder (VAE), to obtain the estimated sources.

## Figures and Tables

**Figure 1 sensors-19-04541-f001:**
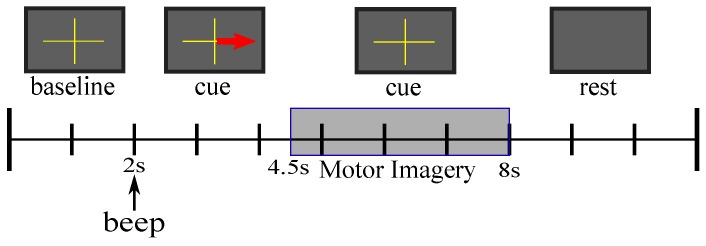
Extraction of MI for each trial.

**Figure 2 sensors-19-04541-f002:**
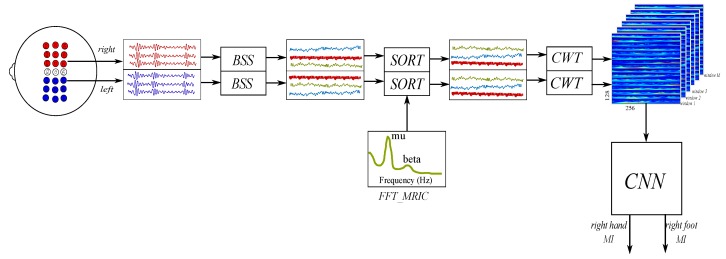
Proposed methodology. The left and right channels are prepossessed using a BSS algorithm, the MRIC sorts the estimated sources, in the CWT stage the images for each time window are obtained, finally the CNN separates the classes.

**Figure 3 sensors-19-04541-f003:**
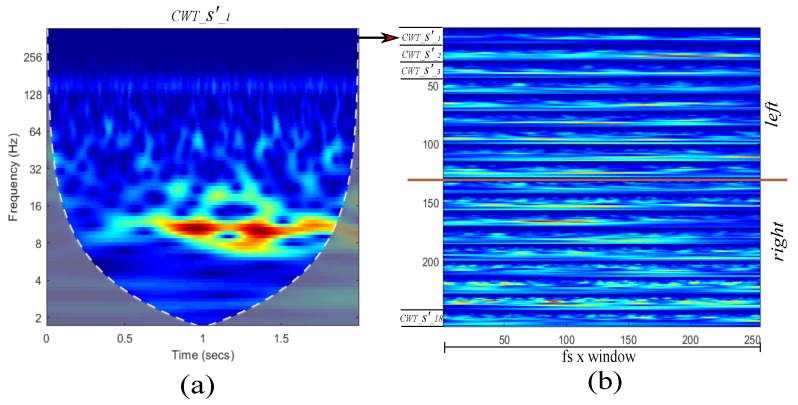
CWT maps for (**a**) one estimated source; (**b**) CWT stacked maps for left and right estimated sources.

**Figure 4 sensors-19-04541-f004:**
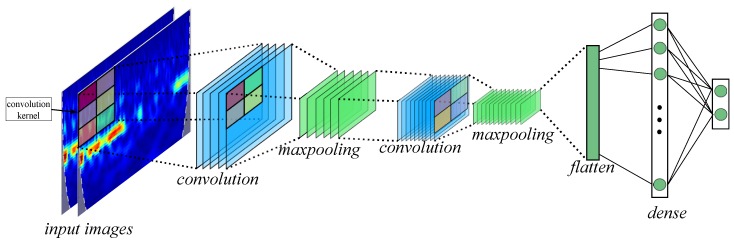
Scheme of CNN architecture used. The CWT input images are pass through two convolutional layers with respective max-pooling. The matrices are flattened and passed by a dense layer.

**Figure 5 sensors-19-04541-f005:**
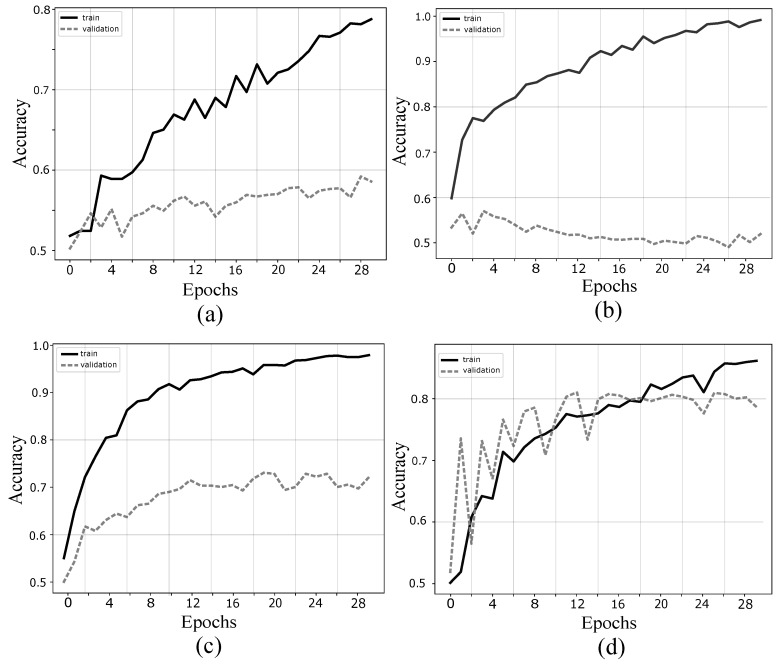
Train and test validation behaviour for subject *aa* in cases. (**a**) without BSS; (**b**) with no sorted fastICA; (**c**) with sorted SOBIRO; and (**d**) with sorted fastICA (30 epochs). with the sorted HOS fastICA, both the training data and the validation data achieve values higher than 0.8, reducing the phenomenon of overfitting in comparison to the other cases.

**Figure 6 sensors-19-04541-f006:**
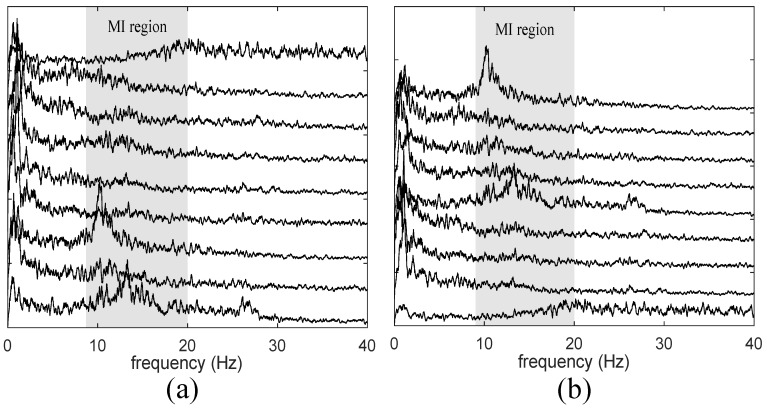
Analysis spectral components: (**a**) before sort; (**b**) after sort. The components with more MRIC frequencies are placed at the top after sorting.

**Figure 7 sensors-19-04541-f007:**
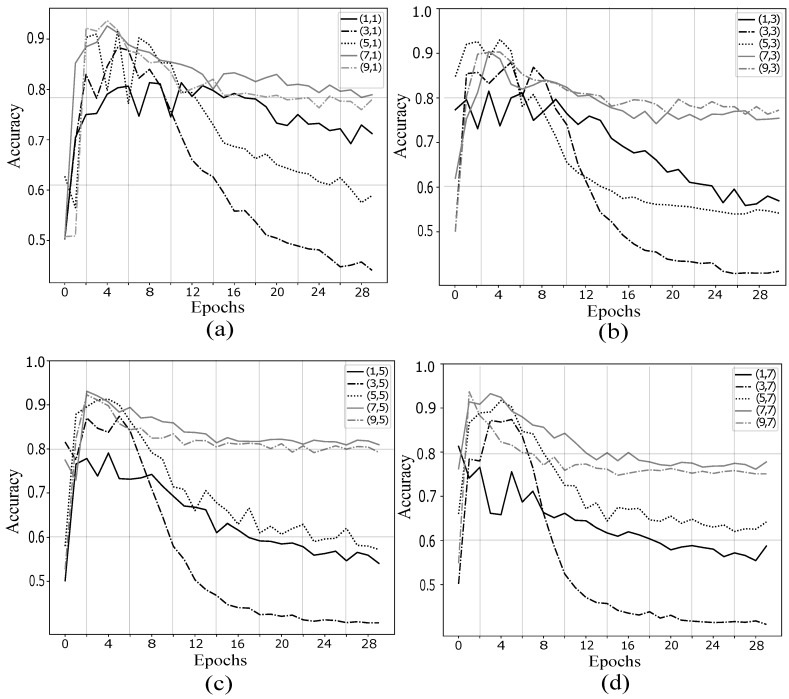
Analysis of kernel size in first convolutional layer: (**a**) (i,1); (**b**) (i,3); (**c**) (i,5); and (**d**) (i,7). The kernel size (7,5) in (**c**) presented less overfitting and major classification accuracy.

**Figure 8 sensors-19-04541-f008:**
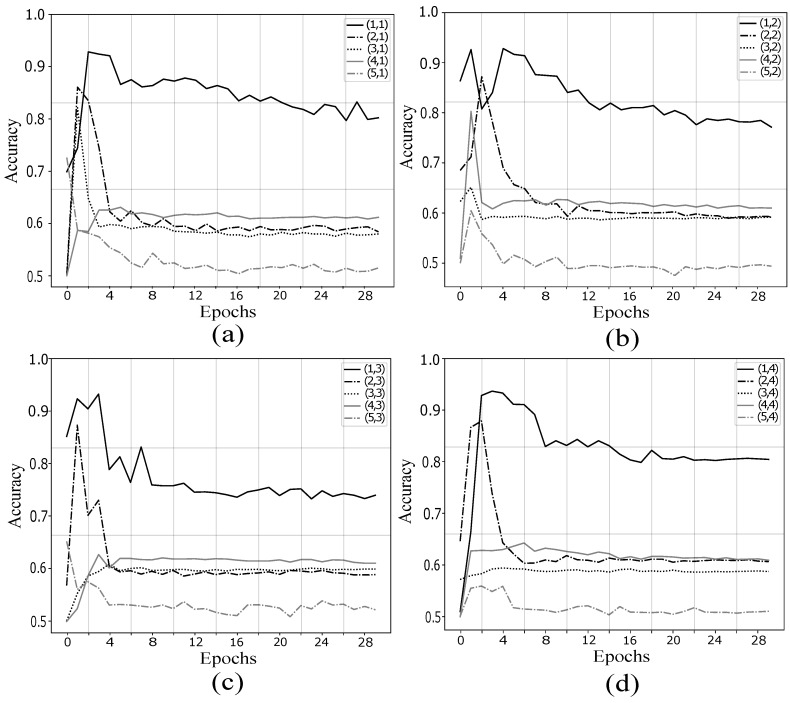
Analysis of kernel size in second convolutional layer: (**a**) (j,1); (**b**) (j,2); (**c**) (j,3); and (**d**) (j,4). The kernel size (1,1) in (**a**) presented less overfitting and major classification accuracy. The only *y*-axis where the CNN can achieve a stable accuracy throughout the epochs is 1.

**Table 1 sensors-19-04541-t001:** CNN modified architecture inspired from [32].

Layer	Operation	Kernel	Stride	Output Shape
1	Conv2D →250	(3,1)	(2,2)	(63,256,250)
	Activation→ReLU			(63,256,250)
	Max-pooling→(4,4)			(15,64,250)
2	Conv2D →150	(1,2)	(1,1)	(15,63,150)
	Activation→ReLU			(15,63,150)
	Max-pooling→(3,3)			(5,21,150)
3	Flatten			(15750)
	Dense→2048			(2048)
	Activation→ReLU			(2048)
	Dropout→0.4			(2048)
4	Dense→2			(2)
	Activation→softmax			(2)

**Table 2 sensors-19-04541-t002:** 10-fold cross validation accuracy.

Subject	Accuracy											
	k = 1	k = 2	k = 3	k = 4	k = 5	k = 6	k = 7	k = 8	k = 9	k = 10	Average	Std
subject aa	94.79	100.00	96.87	89.58	100.00	100.00	100.00	98.95	98.95	98.95	97.81	3.34
subject al	91.66	94.79	94.79	98.95	85.41	87.50	97.91	100.00	98.95	94.79	94.47	4.96
subject av	95.83	97.91	100.00	98.95	97.91	88.54	68.75	100.00	100.00	100.00	94.78	9.79
subject aw	98.75	92.50	95.00	99.37	99.37	91.87	100.00	98.75	76.87	88.12	94.06	7.26
subject ay	85.41	97.91	85.41	79.16	96.87	95.83	95.83	100.00	96.87	88.54	92.18	6.98
Average											**94.66**	6.46

**Table 3 sensors-19-04541-t003:** Comparison with other works using the IVa of BCI competition III dataset.

Author	Method	Classifier	Accuracy (%)	Year
Lu et al.	R-CSP with aggregation	R-CSP	83.90	2010
Siuly et al.	CT	LS-SVM	88.32	2011
Zhang et al.	Z-score	LDA	81.10	2013
Siuly et al.	OA	NB	96.36	2016
Kevric et al.	MSPCA, WPD, HOS	k-NN	92.80	2017
Taran et al.	TQWT	LS-SVM	96.89	2018
Proposed	sorted-fastICA-CWT	CNN	**94.66**	2019

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
