# Peer review of "A New Approach for Motor Imagery Classification Based on Sorted Blind Source Separation, Continuous Wavelet Transform, and Convolutional Neural Network"

_sensors, 2019, doi:10.3390/s19204541_

Round 1

Reviewer 1 Report

The work is well written and interesting. However, it is necessary to make some observations.

- In the Methodology section it is not reported which wavelet was used to calculate the CWT. This is important because the performance depends on which wavelet used.

- Considering that the amount of data is small, why deep learning is used?

- The impression of the work is that "computationally heavy" tools were applied (such as BBS, CWT and CNN) and are not reflected in the results obtained, as shown in table 3.

Author Response

Reviewer 1:

In the methodology section it is not reported which wavelet was used to calculate the CWT. This is important because the performance depends on which wavelet used.

Response: Thank you for the observation. The mother wavelet is now described in the “proposed approach” subsection in line 148. “...The CWT is obtained using the analytic Morse wavelet. Generalized Morse wavelets are a family of analytic wavelets. Analytic wavelets are complex-valued wavelets whose Fourier transforms are supported only on the positive real axis. They are useful for analyzing modulated signals, which are signals with time-varying amplitude and frequency [44]...”

Considering the amount of data small, why deep learning is used?

Response: We also considered this question during the development of our work. The proposed structure is not really that deep (2 hidden layers) in comparison with other computer vision applications. However, the term “deep learning” has been preserved in CNN used in MI in which datasets of similar size are employed [26, 27, 29, 30, 31, 32].

The impression of the work is that “computationally heavy” tools were applied (such as BSS, CWT and CNN) and are not reflected in the results obtained, as shown in table 3.

Response: Initially the training stage of CNN can be “computationally heavy”, but when the parameters have been properly adjusted the CNN is fast (0.05 seconds in our experiment). On the other hand, the window size necessary to obtain informative MI features comprises a time-lapse greater than the total time of processing. For example for a single trial of 3.5 seconds of the 18 channels, the BSS processing time is  1.19 seconds, for CWT is 0.96 seconds, i.e. 2.15 seconds overall.

Reviewer 2 Report

Paper "A New Approach for Motor Imagery Classification based on Sorted Blind Source Separation, Continuous Wavelet Transform, and Convolutional Neural Network" concerns the BCI based on electroencephalographic signals which are obtained from the dataset IVa from the BCI competition III. In this work, Authors use the combination of the Blind-Source-Separation method, the continuous wavelet transform and as a classifier the convolutional neural network. A spectral correlation with a movement related independent component is applied to sort the sources by reducing the spatial variance. The Authors are aware that the analyzed database of signals is small, so they propose a modified approach to its analysis. The paper deals with a very interesting topic of the brain-computer interface research, contains interesting algorithms, but when describing the applied methods, the authors make some mental shortcuts to where the transparency of work suffers. I am not an English specialist, but the work needs to be checked by a native speaker.

During studying the paper I noticed few things which did not have a good explanation and they have impact on quality of the paper. Here are a few of them.
1. line 78-81: Authors use the HOS fastICA algorithm, but there is no explanation why this algorithm is better than that based on Second-Order Statistics.
2. The equation (3) the explanation of H(s) is missing.
3. The equation (4), there is J(yi) but what is yi?
4. line 119: the dataset VIa from the BCI competition III ... it should be IVa.
5. line 119: there is no reference for this dataset and no information about accessibility (some link).
6. line 161, subsection 4.1 Validation of proposed method: with sorted SOBIRO? what is this shortcut? What is the relationship with BSS algorithm?
7. Problems with clarity and readability of drawings, figures 5, 7, 8 descriptions of axis and legend.
8. line 209-211, this sentence is complicated and should be changed.
9. line 241-242, the Authors mention different interferences that can be observed in the tested signals, but can they give specific examples (based on the analyzed signals), where such interferences occurred and the proposed method of analysis was resistant to this type of interferences

Author Response

Reviewer 2:

The paper deals with a very interesting topic of the brain computer interface research, contains interesting algorithms, but when describing the applied methods, the authors make some mental shortcuts to where the transparency of work suffers. I am not an English specialist, but the work needs to be checked by a native speaker. During studying the paper, I noticed few things which did not have a good explanation and they have impact on quality of the paper. Here are a few of them.

Thank you for your suggestion. A native speaker has reviewed the latest version of the manuscript.

Line 78-81: authors use the HOS fastICA algorithm, but there is no explanation why this algorithm is better than based on Second-Order Statistics.

Response: Based on your suggestion, the explanation and references are included in the paragraph. “...SOS algorithms searches the independence of signals based on the criterion of correlation between them. However, uncorrelatedness does not imply independence in all cases. In contrast HOS algorithms are based on the assumption that independent sources have non-Gaussian PDF, while the mixtures of sources before separation present a Gaussian  PDF, which is valid for EEG signals [45]…”.

The equation (3) the explanation of H(S) is missing.

Response: Thank you for the observation. The definition of random variable was improved and now its explanation is included.  “...which is a measure of the entropy of a random variable H(y) with respect to the entropy of a Gaussian distribution variable Hg(y)...”.

The equation (4), there is J(yi) but what is yi?

Response: Thank you for helping us find some errors, we are sorry because of that. The notation was standardized. Now yis the component of a random variable Y.

Line 119: the dataset VIa from the BCI competition III…it should be IV

Response: Thank you for helping us find errors in the redaction. The typing error was corrected.

Line 119: there is no reference for this dataset and no information about accessibility (some link).

Response:  We have included the link in reference [41].

Line 161, subsection 4.1 validation of proposed method: with sorted SOBIRO? What is this shortcut? What is the relationship with BSS algorithm?

Response: This is an important observation. In this section, the performance of a SOS algorithm: the Robust Second Order Blind Identification (SOBIRO) is compared with a HOS algorithm (fastICA). The description of SOBIRO is included in the mentioned subsection 4.1. validation of proposed method.“...The first experiment was carried out to compare the performance between two BSS approaches: SOS and HOS. To evaluate the proposed methodology, an SOS called Robust Second Order Blind Identification (SOBIRO), and the previously mentioned fastICA, belonging to the family of HOS algorithms were tested in one subject…”

Problems with clarity and readability of drawings figures 5,7,8 descriptions of axis and legend.

Response: The axis and legend of Figures 5,7, and 8 were improved.

Line 209-211, this sentence is complicated and should be changed.

Response: Thank you, we appreciate the help in the manuscript grammar improvement. Taking this point into account, the sentence was rewritten.

Line 241-242: the authors mention different interferences that can be observed in the tested signals, but can they give specific examples (based on the analyzed signals), where such interferences occurred and the proposed method of analysis was resistant to this type of interferences.

Response: The mentioned interferences are reported in the literature related to EEG source separation [16]. In Figure 6 (b), the top sources contain major mu and beta rhythms, while the bottom sources contain major components of electrical noise. This means that in the CWT maps, the undesired signals will be at the bottom of each image, and the convolutional operations learn the relevant patterns by discarding the aforementioned interferences.

Reviewer 3 Report

This study describes a signal processing method for a motor imaginary classification task. The method combines deep learning neural network with fastICA and continuous wavelet transform. The open-source dataset is used and comparison of results with other 6 methods proposed during previous years is provided. The results presented in terms of classification accuracy are not better than published by other authors in 2016 and 2018. The article is not technically sound. It is not clear how the problem of large data set needed for training such a neural network was solved. The presentation lacks clarity. Important details are consistently missing. The authors claim improved classification accuracy (94.21%), however, they do not provide any clues for how this improvement was obtained.

Major and minor points in the article which needs clarification and refinement:

A major point in the article which needs clarification and refinement is the question of what novelty is introduced by the authors to the field? This information must be explicitly stated in the Introduction. The Introduction is poorly written. There is no explanation for the motor imaginary concept. What does it mean “IM-based” in this sentence “which makes signal processing for IM-based BCI more complex”? What frequency range has mu or sensorimotor rhythm? What kind of bandpass filter was used - FIR or IIR? Why these cut-offs 0.5 and 90 Hz were selected? The description of the dataset is not adequate: how were data collected? How were data arranged in training and testing data sets? It is difficult to understand “These sources are sorted correlating the their spectral components, MRIC which contain mu and beta frequencies”. Please correct this sentence. I don’t understand this sentence part“…with a window of length 1.0…”. Are variables a in (5) and (6) the same? Equation (7) is not right. The captions of the figures are not informative. Where is "red line" and "80%" mentioned in “In the graphs it is possible to observe that in case a) without preprocessing BSS stage, the maximum accuracy value for train (red line) is near to 80%,…”? What does it mean “CWT reduces the uncertainty principle”? Figures 7-8 are of poor quality.

Author Response

Reviewer 3

This study describes a signal processing method for a motor imaginary classification task. The method combines deep learning neural network with fastICA and continuous wavelet transform. The open-source dataset is used and comparison of results with other 6 methods proposed during previous years is provided. The results presented in terms of classification accuracy are not better than published by other authors in 2016 and 2018. The article is not technically sound. It is not clear how the problem of large data set needed for training such a neural network was solved. The presentation lacks clarity. Important details are consistently missing. The authors claim improved classification accuracy (94.21%), however, they do not provide any clues for how this improvement was obtained.

Major and minor points in the article which needs clarification and refinement:

A major point in the article which needs clarification and refinement is the question of what novelty is introduced by the authors to the field? This information must be explicitly stated in the Introduction. The Introduction is poorly written.

Response: Thank you for this important observation. As you suggest, now in the introduction is included: “...The major contribution of the present work is the use of BSS instead of the widely used CSP. Even though this has worked well for MI-BCI based, these spatial filters require prior information of the classes to be separated [7]. In addition, BSS is an unsupervised approach that does not require prior information about the classes [15]. The problem of large datasets needed for training is minimized using the MRIC criterion to sort the estimated sources…”. In Figure 5 is shown the comparison with and without the use of BSS, as well as the influence of sort the components in terms of accuracy performance.

There is no explanation for the motor imaginary concept.

Response: Thank you for the suggestion. Motor Imagery concept in the introduction is now included: “... or endogenous potentials, where Motor Imagery (MI) widely used in BCI applications is the dynamic state where a subject evokes a movement or gesture. The event related phenomena represent frequency-specific changes in the ongoing EEG activity and may consist, in general terms, of either decreases or increases of power in given frequency bands…”.

What does it mean “IM-based” in this sentence “which makes signal processing for IM-based BCI more complex”

Response: We are sorry for the mistake. The idea in this sentence refers to the fact that feature extraction and classification in MI-BCI is more difficult in comparison with evoked potentials. This sentence was rewritten as: “...Nonetheless, MI depends on the ability to control the electrophysiological activity, which makes feature extraction and classification for MI-BCI based system, more difficult than for exogenous response…”.

What frequency range has mu or sensorimotor rhythm?

            Response: Thanks!, Sensorimotor mu rhythm has the same frequency as alpha rhythm (8-13 Hz) but located in sensorimotor cortex. This information is now included in: “...Another important frequency for applications in BCI is the μ or sensorimotor rhythm, with the same frequency bands as α , but located in the motor cortex instead of the visual cortex…”.

What kind of bandpass filter was used - FIR or IIR?

Response: Thank you, an IIR passband filter was employed. This has been included in the latest version of the manuscript as “...The two clusters are filtered using a second-order IIR Butterworth, set as band-pass filter…”.

Why these cut-offs 0.5 and 90 Hz were selected?

Response: According to the literature, the associated frequency for MI is between 8 and 30 Hz [8-10]. Therefore, the cut-off frequency selection was made so that these frequencies would not be attenuated by the filter. In addition, this cut-off helps to reduce the effect of some movement artifacts generated such as blinks, muscle movement, cardiac activity, and electrical noise above 100 Hz. The explanation was included in the dataset from subsection 3.1. It is included in: “...To preserve the relevant information for MI, the selected cut-off frequencies were adjusted between 0.5 and 90 Hz…”.

The description of the dataset is not adequate: how were the data collected?

Response: Based on your comment, the description of the dataset was improved in subsection 3.1. “...The recording was made using BrainAmp amplifiers and a 128 channel Ag/AgCl electrode cap from ECI. 118 EEG channels were measured at positions of the extended international 10/20-system. Signals were digitized at 1000 Hz with 16 bit (0.1 uV) accuracy…”.

How were data arranged in training and testing data sets?

Response: The datasets were divided into 50% for train and 50% for test such as in [32]. To generate the k-fold cross validation, only the test part was subdivided in k=10 fragments . This is described in subsection 4.2 comparison with other methods.

It is difficult to understand “These sources are sorted correlating their spectral components, MRIC which contain mu and beta frequencies”. Please correct this sentence.

Response:  The sentence was rewritten such as: “...These sources were sorted using as criterion the correlation between their spectral components and the MRIC…”

I don’t understand this sentence part“…with a window of length 1.0…”.

Response: The sentence was rewritten as: “...The window size of each CWT is 1.0 seconds, each window is computed using steps of 0.25 seconds...”

Are variables a in (5) and (6) the same?

Response: Thank you for this observation. The variable a in (5) and (6) are not the same. In (5) a is the scale factor in the wavelet operation, while in (6) it refers to the convolutional outputs inside an activation function. To avoid confusion, the variable a in (6) was changed to c.

Equation (7) is not right.

Response: Thank you for helping us to find errors in the equations. The equation 7 was corrected.

The captions of the figures are not informative.

Response: Attending to your comment. All captions were improved.

Where is "red line" and "80%" mentioned in “In the graphs it is possible to observe that in case a) without preprocessing BSS stage, the maximum accuracy value for train (red line) is near to 80%,…”?

Response: Thank you for this observation. The mistake was corrected.

What does it mean “CWT reduces the uncertainty principle”?

Response: According to literature [39], CWT reduces the uncertainty principle inherent in STFT, i.e. when the window size in the time domain is small, there are a good time resolution, but there is not enough information to obtain a good frequency resolution. On the other hand, when the window size gets larger, the frequential information increase, but the uncertainty in time is bigger too. The multiresolution analysis in CWT helps to reduce this effect.

Figures 7-8 are of poor quality. 

Response: Thank you for the observation. Figures 7 and 8 were improved.

Round 2

Reviewer 3 Report

The presentation of the method was improved. 

However, please reformulate your conclusions which state "...CWT reduces the uncertainty principle thanks to the multi-resolution analysis". CWT does not change the uncertainty principle. It is a law of Nature.  

Author Response

Response: Thank you for the observation. The sentence was rewritten as: “…CWT performs a multi-resolution analysis”.